# Parapoxvirus Interleukin-10 Homologues Vary in Their Receptor Binding, Anti-Inflammatory, and Stimulatory Activities

**DOI:** 10.3390/pathogens11050507

**Published:** 2022-04-24

**Authors:** Amreen Naqash, Gabriella Stuart, Roslyn Kemp, Lyn Wise

**Affiliations:** 1Department of Pharmacology and Toxicology, School of Biomedical Sciences, University of Otago, Dunedin 9054, New Zealand; amreen.naqash@postgrad.otago.ac.nz (A.N.); gabriella.stuart@otago.ac.nz (G.S.); 2Department of Microbiology and Immunology, School of Biomedical Sciences, University of Otago, Dunedin 9054, New Zealand; roslyn.kemp@otago.ac.nz

**Keywords:** parapoxvirus, virus, IL-10, IL-10 receptor, inflammation, immune stimulation, immune suppression, monocyte, mast cell

## Abstract

Homologues of interleukin (IL)-10, a pleiotropic immunomodulatory cytokine, have been identified in the *Parapoxvirus* genus. The first identified, *Orf virus* (ORFV) IL-10, greatly enhanced infection of its host, exhibiting immune modulatory effects equivalent to human IL-10. IL-10-like genes were then identified in *Bovine papular stomatitis virus* (BPSV), *Pseudocowpox virus* (PCPV), *Red deerpox virus* (RDPV) and *Grey sealpox virus* (GSPV). This study aimed to produce and characterise recombinant parapoxvirus IL-10s, then quantitatively compare their receptor binding and immunomodulatory activities. Recombinant IL-10s were expressed, purified, then characterised using bioinformatic, biochemical and enzymatic analyses. Anti-inflammatory effects were assessed in lipoteichoic acid-activated THP-1 monocytes, and stimulatory effects in MC/9 mast cells. IL-10 receptor (IL-10R)1 binding was detected in a competitive displacement assay. BPSV IL-10 inhibited production of monocyte chemoattractant protein (MCP)-1, IL-8 and IL-1β, induced mast cell proliferation, and bound IL-10R1 similarly to ORFV IL-10. PCPV IL-10 showed reduced MCP-1 inhibition, mast cell proliferation, and IL-10R1 binding. RDPV IL-10 displayed reduced inhibition of IL-8 and MCP-1 production. GSPV IL-10 showed limited inhibition of IL-1β production and stimulation of mast cell proliferation. These findings provide valuable insight into IL-10 receptor interactions, and suggest that the parapoxvirus IL-10s play similar pathogenic roles during infection of their hosts.

## 1. Introduction

Interleukin-10 (IL-10) is a pleiotropic cytokine secreted by immune cells that plays a crucial role in homeostasis [1,2]. IL-10 can act as both an immunosuppressant and immunostimulant, reducing the production of proinflammatory mediators from macrophages and antigen presentation to T cells by dendritic cells (DC), yet activating mast cell, B cell and CD8 T cell proliferation, and increasing interferon γ production and cytotoxic activity. The importance of IL-10 is highlighted by the fact its dysregulation underlies many human diseases, including psoriasis, Crohn’s disease, systemic lupus erythematosus, and asthma [3,4,5,6].

IL-10 is a secreted homodimer consisting of two monomers, each comprising six α-helices. IL-10 engages a tetrameric receptor complex composed of two class II cytokine receptor family subunits: IL-10R1 and IL-10R2 [7,8]. IL-10R2 is widely expressed, while IL-10R1 is produced at high levels in macrophages, and lower levels in other leukocytes. IL-10 binding activates Janus kinase (JAK) 1 and tyrosine kinase (TYK) 2, which leads to phosphorylation of IL-10R1 and recruitment of members of the signal transducers and activators of transcription (STAT) family. Phosphorylation of STAT results in its translocation into the nucleus, where transcription of anti-inflammatory effector proteins is initiated [9]. STAT3 is the primary STAT activated by IL-10 [10], although at higher ligand concentrations IL-10 has also been shown to activate STAT1 and STAT5 [11]. In addition to the JAK1/STAT3 pathway, IL-10 has been shown to activate AMPK and signalling through the PI3K/Akt/GSK3β pathway, leading to transcriptional and mTORC1 activity [12,13].

Viruses, in particular the *Herpesviridae* and *Poxviridae*, are capable of capturing genes from the genomes of their hosts, which they then use to enhance their capacity to replicate, disseminate, and/or persist in otherwise immunocompetent individuals [14,15,16]. Many of these viruses express homologues of human (h)IL-10 which exploit IL-10 receptor signalling to shield infected cells or tissue from the inflammatory and antiviral activity of the immune system [17]. These viral IL-10s have provided greater understanding of hIL-10, as they differ in receptor affinity and functionality across different cell types. The first IL-10-like gene identified was found in the genome of the *Epstein–Barr virus* (EBV) [18]. Subsequently, IL-10-like genes were detected in the genomes of *Human Cytomegalovirus* (HCMV) and *Orf virus* (ORFV) [19,20].

Orf virus is the prototype member of the *Parapoxvirus* genus within the Poxviridae [14]. Other members of the parapoxviruses include *Bovine popular stomatitis virus* (BPSV), *Red deerpox virus* (RDPV), *Pseudocowpox virus* (PCPV), and *Grey sealpox virus* (GSPV) [21]. These parapoxviruses are all zoonotic pathogens that cause cutaneous pustular lesions [22,23,24,25,26,27,28,29]. ORFV (strain NZ2) was found to encode an IL-10-like protein with high amino acid similarity to the IL-10s from sheep (80%), humans (67%), and EBV (63%) [30]. Functional characterisation revealed that the ORFV IL-10 shares most biological activities of hIL-10. In in vitro settings, ORFV IL-10 inhibited cytokine production in stimulated human, murine and ovine monocytes, reduced costimulatory molecule expression and antigen presentation in DC, and stimulated mast cell and thymocyte proliferation [30,31,32,33]. In a murine model, ORFV IL-10 inhibited trafficking of monocytes, DC and mast cells to the site of skin inflammation [34]. Deletion of the IL-10-like gene also severely attenuated ORFV, reducing lesion size and severity during infection and reinfection of sheep skin [35]. Recently, IL-10-like genes have been reported in BPSV, RDPV, PCPV, and GSPV [36,37,38,39]. Given the similarities in parapoxvirus pathology and the important virulence role that ORFV IL-10 plays during host infection, we hypothesised that the newly reported parapoxvirus IL-10s would exert similar biological effects to the ORFV IL-10. In this study, we aimed to produce, purify and characterise recombinant BPSV, RDPV, PCPV, and GSPV IL-10s, then quantitatively compare their receptor binding and immunomodulatory activities to hIL-10 and ORFV IL-10.

## 2. Results

### 2.1. Parapxvirus IL-10s Differ in Amino Acid Sequence, Molecular Weight and Glycosylation State

Gene sequences for the BPSV, RDPV, PCPV and GSPV IL-10s were sourced from GenBank, then signal peptide predictions were made to estimate the length of their mature polypeptides (Appendix A). It was estimated that the parapoxvirus IL-10s would range from 185 to 204 amino acids, resulting in molecular weights from 17 kDa to 21 kDa, with the RDPV and PCPV IL-10s corresponding to the smallest and largest proteins, respectively (Appendix A).

The mature polypeptides of the parapoxvirus IL-10s were aligned with that of hIL-10 and retention of secondary structures assessed. The alignment indicated that each parapoxvirus IL-10 would comprise of six α-helices, as with the hIL-10 monomer (Figure 1). High sequence conservation was evident between the PPV IL-10s and hIL-10 within α-helices B, C, and F, and the B-C loop. Although α-helices A and D and loops A-B, D-E, and E-F showed similarity between hIL-10 and the ORFV, PCPV and BPSV IL-10s, the RDPV and GSPV IL-10s were divergent in these regions. All PPV IL-10s were predicted to contain N-terminal extensions relative to hIL-10, with the longest found in the GSPV IL-10. The N-terminal cysteine residue adjacent to α-helix A that is involved in disulfide bond formation within hIL-10 was conserved between hIL-10 and the ORFV, BPSV, PCPV and GSPV IL-10s, but was missing in the RDPV IL-10. The parapoxvirus IL-10s did, however, share an N-terminal cysteine residue not present in human IL-10. PCPV IL-10 had a C-D loop extension not present in any other IL-10, while RDPV IL-10 had an insertion in α-helix E.

The parapoxvirus IL-10 sequences were next analysed for the presence of *N*-linked glycosylation sites. All IL-10s were predicted to contain an *N*-glycan within the D-E loop (Figure 1); however, the motif differed in the RDPV IL-10 (NTT vs. NKS in the other IL-10s). Additional N-linked glycosylation sites were predicted in or adjacent to α-helix A for GSPV IL-10 (NSS) and PCPV IL-10 (NHS), respectively.

Recombinant human and PPV IL-10s, tagged at the C terminus with the FLAG octapeptide, were expressed and purified, then identified by Western blotting following SDS-PAGE gel electrophoresis (Figure 2 and Appendix A). In the presence of reducing agent, protein bands were observed at ≈19 kDa for hIL-10 and the ORFV, PCPV, and BPSV IL-10s (Figure 2a,b). The molecular weights were consistent with the monomeric sizes previously reported for hIL-10 and ORFV IL-10 [30,40], and that predicted for the PCPV and BPSV IL-10s (Appendix A). However, the bands corresponding to the RDPV and GSPV IL-10s were larger than predicted at 21 kDa (Figure 2a) and 25 kDa (Figure 2b), respectively. For the human and RDPV IL-10s, bands that were smaller than the monomers were also observed (Figure 2a,b).

Without reducing agent, bands of increased size indicative of dimer formation (≈34 kDa) were observed in low abundance for hIL-10 and the ORFV and PCPV IL-10s (Figure 2a). This indicates that, like hIL-10 [40,41], these viral IL-10s exist as unstable non-covalent dimers. Larger bands were also observed for the PCPV IL-10 (Figure 2a and Appendix A), which were suggestive of protein aggregation. No protein bands greater than the monomeric size were observed for the RDPV, GSPV or BPSV IL-10s (Figure 2a,b).

As the RDPV and GSPV IL-10 protein bands did not correspond to their predicted molecular weights, and were predicted to contain *N*-glycan, all IL-10s were subjected to *N*-glycosidase F digestion prior to Western blot analysis (Figure 3 and Appendix A) [42]. No change in band size was observed following digestion of hIL-10 or the ORFV, BPSV and PCPV IL-10s (Figure 3a–c). However, the size of the GSPV IL-10 band decreased from 25 kDa to 20 kDa (Figure 3c), while that of the RDPV IL-10 decreased from 21 kDa to 17 kDa (Figure 3b).

These bioinformatic and protein analyses show that the parapoxvirus IL-10s exhibit differences in their amino acid sequence, molecular weight, and glycosylation state, which may impact on their structure and function.

### 2.2. Parapoxvirus IL-10s Vary in Their Inhibition of Pro-Inflammatory Mediator Production in Stimulated Human Monocytes

To assess whether differences between the parapoxvirus IL-10s affect their function, their anti-inflammatory effects were compared to hIL-10 in human THP-1 monocytes stimulated with the toll-like receptor 2 agonist lipoteichoic acid (LTA) [43]. Changes in production of the chemokines, IL-8 and monocyte chemoattractant protein (MCP)-1, and the cytokine, IL-1β, were assessed by ELISA at 24, 24 and 48 h, respectively. In response to LTA stimulation, THP-1 cells showed an increase in the production of IL-8, MCP-1, and IL-1β compared to unstimulated controls (12-, 9- and 70-fold, respectively, Appendix A). Reduced IL-8, MCP-1, and IL-1β levels were observed with increasing concentrations of hIL-10 and ORFV IL-10 (Figure 4a–c). Treatment with BPSV IL-10 also inhibited IL-8, MCP-1, and IL-1β production in a concentration-dependent manner (Figure 4d–f). However, PCPV IL-10 treatment resulted in concentration-dependent inhibition of IL-8 and IL-1β (Figure 4d,e), but not MCP-1 (Figure 4f). By contrast, RDPV IL-10 treatment inhibited production of IL-1β (Figure 4i) to a greater extent than MCP-1 and IL-8 (Figure 4g,h). Treatment with GSPV IL-10 showed minimal inhibition of pro-inflammatory chemokine and cytokine production (Figure 4g–i).

The potency and efficacy of the parapoxvirus IL-10s at inhibiting inflammatory mediator production were compared to hIL-10 (Table 1). The ORFV IL-10 was significantly more potent at inhibiting IL-8 production than hIL-10 (Table 1, *p* ≤ 0.05), while the BPSV, PCPV and PRVD IL-10s showed similar efficacy. The negative log10 (p) of the IC50 for GSPV IL-10 was unable to be determined for the inhibition of this chemokine. The ORFV and BPSV IL-10s showed the greatest efficacy, followed by PCPV IL-10 then hIL-10, with RDPV and GSPV IL-10s showing significantly reduced efficacy (Table 1, *p* ≤ 0.05). The ORFV and RDPV IL-10s were equivalent to hIL-10 in their potency at reducing MCP-1 production, while the BPSV IL-10 showed significantly reduced potency (Table 1, *p* ≤ 0.05). The pIC50 values for the PCPV and GSPV IL-10s suppressing this chemokine were not determinable. The ORFV and BPSV IL-10s were the most efficacious inhibitors of MCP-1 production, followed by hIL-10, RDPV, and PCPV IL-10, with GSPV IL-10 showing significantly reduced efficacy (Table 1, *p* ≤ 0.05). All parapoxvirus IL-10s showed similar potency to hIL-10 at reducing IL-1β production (Table 1). The PCPV IL-10 was the most efficacious inhibitor of IL-1β production, with hIL-10 and the ORFV, BPSV, and RDPV IL-10s showing similar efficacy, and GSPV IL-10 being the least efficacious (Table 1).

These findings demonstrate that the sequence variation between parapoxvirus IL-10s translates to differences in their efficacy as anti-inflammatories. The ORFV and BPSV IL-10s showed greater or equivalent anti-inflammatory action to hIL-10, while the PCPV and RDPV IL-10s were biased towards the inhibition of different pro-inflammatory mediators, and the least active anti-inflammatory protein was GSPV IL-10.

### 2.3. Parapoxvirus IL-10s Vary in Their Stimulation of Murine Mast Cell Proliferation

Next, the proliferative effects of the parapoxvirus IL-10s were compared to hIL-10 in murine MC/9 mast cells co-cultured with IL-4 and IL-3 [32]. Changes in cell metabolism were assessed through tetrazolium dye reduction with conversion to cell number through use of a standard curve. MC/9 cell number increased 2.0-, 2.2-, 3.2- and 6.9-fold in response to stimulation with IL-4, IL-3, IL-4 and IL-3, and IL-4, IL-3 and hIL-10, respectively (Appendix A). The extent of MC/9 cell proliferation in response to each IL-10 was then expressed as a percentage of the maximal response obtained with hIL-10. MC/9 cell proliferation was higher with increasing concentrations of ORFV IL-10 than with hIL-10 (Figure 5a). Incubation with BPSV IL-10 increased cell proliferation in a concentration-dependent manner (Figure 5b); however, PCPV IL-10 exposure did not increase cell proliferation to the same extent. Similarly, RDPV IL-10 exposure induced a concentration-dependent increase on cell proliferation (Figure 5c), while incubation with GSPV IL-10 had a modest effect.

The potency and efficacy of the parapoxvirus IL-10s at stimulating mast cell proliferation were compared to hIL-10 (Table 2). All parapoxvirus IL-10s showed similar potency to hIL-10 at inducing cell proliferation (Table 2). However, the ORFV, BPSV, and RDPV IL-10s showed significantly greater efficacy than hIL-10, while the PCPV IL-10 and GSPV IL-10s showed significantly reduced efficacy (Table 2, *p* ≤ 0.05).

These findings demonstrate that differences in the parapoxvirus IL-10 polypeptides impact their efficacy but not potency as mast cell stimulants. The ORFV, BPSV and RDPV IL-10s were all more efficacious than hIL-10, while the PCPV and GSPV IL-10 showed modest efficacy.

### 2.4. Parapoxvirus IL-10s Vary in Their Binding to Human IL-10R1

To establish whether the differences in function between the parapoxvirus IL-10s relate to their receptor binding capacities, the proteins were tested for their ability to inhibit hIL-10 binding to an Fc chimeric protein containing the extracellular domain of hIL-10R1 using a competitive displacement ELISA [42]. Preincubation of hIL-10 or the ORFV, PCPV, BPSV, and RDPV IL-10 with hIL-10R1 inhibited the receptor from binding to immobilised hIL-10 in a concentration-dependent manner (Figure 6a–c). However, the GSPV IL-10 did not inhibit receptor binding to the same extent (Figure 6c).

The potency and efficacy of the parapoxvirus IL-10s at competitively displacing IL-10R1 were compared to hIL-10 (Table 3). Most parapoxvirus IL-10s showed similar hIL-10R1 binding potency and efficacy to hIL-10 (Table 3), while the pIC50 and Emax for GSPV IL-10 were not measurable (Table 3).

These findings indicate that the reduced capacity of GSPV IL-10 to bind hIL-10R1 is likely responsible for the functional differences observed in anti-inflammatory and stimulatory activity between this and other parapoxvirus IL-10s.

## 3. Discussion

IL-10 is an immunomodulatory cytokine that directs immune cell function and restores homeostasis following inflammation and infection. Viral-encoded IL-10s have been identified and shown to play critical roles in infection of their host, by suppressing inflammation and the induction of anti-viral immune responses. Recent genomic analyses had indicated the presence of novel IL-10-like genes encoded by members of the parapoxviruses. This study, therefore, set out to provide the first biochemical and functional characterisation of the IL-10-like proteins encoded by BPSV, PCPV, RDPV, and GSPV. Despite differences in their amino acid sequence, molecular weight, and glycosylation state, each recombinantly-produced protein exhibited some activity consistent with the human and ORFV IL-10s. The BPSV and PCPV IL-10s, and to a lesser extent the RDPV and GSPV IL-10s, showed anti-inflammatory activity in a stimulated human monocyte cell line, while stimulatory activity in a murine mast cell line was more pronounced for the BPSV and RDPV IL-10s than for the PCPV and GSPV IL-10s. The limited activity of the GSPV IL-10 correlated with a reduced capacity to bind the receptor, hIL-10R1. These findings provide valuable information as to how IL-10 exerts its pleotropic effects and indicate that each parapoxvirus IL-10 will act as a virulence factor, modifying the host response to infection.

Human and ORFV IL-10s have previously been shown to dampen pro-inflammatory mediator production in stimulated monocytes [33], and to stimulate the proliferation of mast cells [32,44]. Here, the BPSV IL-10 showed equivalent potency and efficacy as an anti-inflammatory to the ORFV IL-10. As a mast cell stimulant, the BPSV IL-10 was less efficacious than the ORFV IL-10, but more efficacious than hIL-10. The anti-inflammatory and stimulatory activities of IL-10 have been shown to require engagement of IL-10R1 and IL-10R2 [45]. While no differences were observed between the human, BPSV, and ORFV IL-10s in their capacity to bind hIL-10R1, their interactions with IL-10R2 were not investigated. Mast cells express IL-10R2 at higher abundance than IL-10R1 [46], so the varied efficacy of the IL-10s observed here may relate to their IL-10R2 binding capacity. Structural and mutational investigations have identified residues of hIL-10 critical for binding IL-10R2 [47,48,49,50], the majority of which are found at the N-terminus adjacent to α-helix A. It is within this region that the human, ORFV, and BPSV IL-10s showed the least amino acid similarity, including specific residues of hIL-10 implicated in IL-10R2 binding. For example, Pro 16 is missing in the BPSV IL-10, while Asn 18, Asn 21 and Ser 31 have been replaced with Ser, His and Gly in the BPSV and ORFV IL-10s. Mutational analysis of these residues in the BPSV or ORFV IL-10s would allow for their role in mast cell stimulation to be defined.

Another difference between the BPSV IL-10 and the human and ORFV IL-10s was that only a monomeric form was detected. Previous studies have shown that the dimeric form of hIL-10 is more active than its monomeric form [40,41]. As the BPSV IL-10 was active, this would suggest that it does form a dimer, but that this dimeric form may be less stable than the hIL-10 and ORFV IL-10 dimers under SDS-PAGE condition. This would be surprising as the BPSV IL-10 retains all cysteine residues implicated in hIL-10 dimerisation. It should be noted, however, that we were unable to purify BPSV IL-10s at the high concentrations achieved for other IL-10s, so this may have impacted its stability and/or detection. Higher yields of this protein would enable a more thorough analysis of its association state and structural stability.

The PCPV IL-10 was equivalent as an anti-inflammatory to the ORFV IL-10, but only for two of the three pro-inflammatory mediators examined. The PCPV IL-10 also showed half the efficacy of ORFV IL-10 as a stimulant of mast cell proliferation. Again, these differences could not be attributed to IL-10R1 binding capacity, as the ORFV and PCPV IL-10s showed equivalent binding hIL-10R1. The PCPV IL-10 did differ from the hIL-10 at the N-terminus adjacent to α-helix A, but with Pro 16, Asn 18, Asn 21 and Ser 31 missing or replaced in the same manner as for BPSV IL-10. So, the drastic reduction in the inhibition of MCP-1 production and mast cell proliferation in the PCPV IL-10 is unlikely to result from a reduced affinity for IL-10R2. Like hIL-10 and ORFV IL-10, the PCPV IL-10 showed evidence of dimerisation with conservation of the core cysteine residues, and, despite predictions, was not glycosylated in α-helix A. Unlike the other IL-10s, multimeric forms were detected of the PCPV IL-10, but it is unclear how this may have impacted only selected biological activities of the PCPV IL-10.

An explanation for the reduced immunostimulatory activity and IL-10R1 binding of the PCPV IL-10 may, instead, lie in unique features within its C-D loop. In concert, the A-B and C-D loops of hIL-10 undergo orientational change while interacting with IL-10R1 [51]. Residues within these loops, namely Leu 47 and Ile 87, influence the immunostimulatory properties and IL-10R1 affinity of IL-10, and their importance was identified through comparisons between the human and EBV IL-10s. Cell binding studies revealed that EBV IL-10 has a 1000-fold lower affinity for hIL-10R1 than hIL-10 [52], while functional studies demonstrated its reduced immunostimulatory activity [44,53,54]. EBV IL-10 contains Val 43 and Ala 87, which results in a partially disordered A-B loop that fails to pack into the hydrophobic core of the protein [51]. Mutational studies confirmed that the absence of both or either of Leu 47 and Ile 87 disrupted the molecular core of hIL-10 and reduced contact with the IL-10R1 interface [55]. Substitution of Ile 47 with Ala also abrogated the ability of hIL-10 to induce mast cell proliferation [53], while substitution of Ala 87 with Ile enhanced the immunostimulatory activity of the EBV IL-10.

The PCPV IL-10 contains a 12-amino-acid insertion in the C-D loop, and Ile 87 is substituted with Val. It is possible, therefore, that these changes cause a structural distortion influencing PCPV IL-10’s interaction with IL-10R1. However, in the IL-10R1 displacement assay, the PCVP IL-10 showed only a drop in efficacy not potency. This assay may not be as sensitive as the cell binding assay at detecting changes in IL-10R1 affinity. However, affinities generated in the cell binding assay are likely influenced by IL-10R2 interactions, as this receptor is abundant on most cells [56]. Mutational analysis combined with further receptor binding experiments would establish the impact of the PCPV IL-10’s structural features on its affinity for the IL-10 receptors. However, the findings suggest that the length of the C-D loop and/or substitution of Ile 87 impacts the immunosuppressive properties of PCPV IL-10, and potentially its inhibition of MCP-1 production.

As an anti-inflammatory mediator, the RDPV IL-10 showed reduced activity relative to the ORFV IL-10, with minimal inhibition of chemokine production after 24 h, but effective inhibition of IL-1β production after 48 h. This suggests that RDPV IL-10 may act selectively to regulate the inflammasome, or that the kinetics of its action are slower than with the ORFV IL-10. Further time course and transcriptional analyses would confirm the reason for this disparity. By contrast, the RDPV IL-10 was as effective as ORFV IL-10 at stimulating mast cell proliferation and at binding hIL-10R1. Structurally, RDPV IL-10 differs from hIL-10 to a greater extent than the other parapoxvirus IL-10s. It is truncated at the N-terminus adjacent to α-helix A and, as a consequence, lacks a conserved cysteine residue, which appears to have impacted its dimer formation, stability or detection. Further changes include the loss of Pro 16, Asn 18, and Pro 20, and the substitution of Arg 24 (to Gln), Gln 42 (to Thr), and Arg 107 (to Glu), which may impact its interactions with IL-10R1 and IL-10R2 [50,57,58]. In addition, RDPV IL-10 has a substitution at Ile 87, which has changed to Pro. This protein also contains *N*-glycans, as predicted for the NTT site in α-helix E.

Given the extent of the amino acid changes in the RDPV IL-10, it is surprising that it retains affinity for IL-10R1 and its immunostimulatory activity. However, it has been reported previously that HCMV-IL-10, which shares only 27% amino acid identity with hIL-10, retains an essentially identical affinity for IL-10R1 through subtle changes in the A-B loop [59]. The immunostimulatory activities of hIL-10 have also been enhanced by increasing its affinity for hIL-10R2 through yeast display-based directed evolution [47,48]. Interestingly, RDPV IL-10 contains substitutions at three of these mutations sites, with Asn 92, Lys 99 and Phe 111 changed to Lys, His and Trp, as opposed to Ile, Asn and Leu in the enhanced hIL-10 mutants. Further mutational, structural, and binding analyses would be needed to ascertain which features of the RDPV IL-10 account for its altered balance of anti-inflammatory and immunostimulatory activities.

Compared to the other parapoxvirus IL-10s, the GSPV IL-10 showed the least anti-inflammatory, immunostimulatory, and receptor binding activity. However, some inhibition of IL-1β production and induction of mast cell proliferation was observed after 48 h, indicating that the protein is active to some extent. From a structural perspective, there are many explanations as to why the GSPV IL-10 shows reduced activity. Although the cysteine residue adjacent to the α-helix A is present, no dimer formation was observed for the GSPV IL-10. However, as with BPSV IL-10, this may have resulted from our inability to purify enough protein to achieve high concentrations. The GSPV IL-10 protein also contained *N*-glycans, which were predicted for the NSS site within α-helix A, which is adjacent to critical IL-10R1 binding residues. In addition, substitutions have occurred for Asn 18 (to Ser), Asn 21 (to Tyr), and Gln 42 (to Glu), which may impact IL-10R1 and IL-10R2 binding. Critically, the GSPV IL-10 has substitutions at Leu 43 and Ile 87, which have been changed to Glu and Ala, respectively. As discussed above, these residues are critical for IL-10R1 binding and the immunosuppressive effects of hIL-10. Further mutational analyses are needed to ascertain which of these modifications have the greatest impact on the GSPV IL-10 receptor binding and function. However, it should be noted that this protein may show greater activity and receptor binding in its host, seal species and other pinnipeds. As at the amino acid level, the human and seal IL-10R1 proteins show only 73% identity. The GSPV IL-10 may, therefore, have been more active if our assays utilised host cells and receptors. Many such viral virulence factors that exhibit species specificity, or fail to act in non-host species, have been reported [60,61].

As the parapoxvirus IL-10s differ in their biological responses, and likely receptor interactions, they could provide a valuable tool for dissecting the signalling pathways that control the pleotropic effects of hIL-10. IL-10 engagement of the IL-10 receptor complex activates the JAK/STAT and PI3K/Akt/GSK3β pathways which lead to transcription of anti-inflammatory effector proteins and mTORC1 activity in monocytes/macrophages [11,12]. Many anti-inflammatory response factors, such as SOCS3, Bcl3, Etv3, and SHIP-1, are produced in these cells through STAT3-dependent and -independent mechanisms, and act in concert to suppress NF-κB mediated pro-inflammatory gene expression [9]. Although IL-10 receptor signalling has been defined in monocytes/macrophages, there have been very little investigations into the role of the IL-10 receptors and STAT-3-dependent and -independent signalling pathways in the expression of different anti-inflammatory effector proteins in these or other cell types, nor in the context of IL-10-mediated immune stimulation. The parapoxvirus IL-10s, as naturally-derived IL-10 mutants with differing anti-inflammatory and stimulatory capacities, therefore, could help define how hIL-10 mediates its biological effects across different immune cells. It would be of particular interest to include DC, B cells and CD8 T cells, which were not examined in this study.

Parapoxvirus infection of mammals is confined to the epithelium and oral mucosa [21]. ORFV infects the muzzle of sheep and goats, PCPV and BPSV infect the teats of cattle and mouths of calves, RDPV infection of red deer occurs in the velvet of stags around the face and mouths of fawns, while GSPV infection of the skin and oral mucosa has been reported in seals and sea lions. Zoonotic infections of human skin have also been reported for these parapoxviruses [23,62,63,64,65]. Lesions from ORFV, PCPV, BPSV, and RDPV infections consistently report of dermal oedema and a dense infiltrate of lymphocytes, neutrophils, and macrophages [22,23,24,25,26,27,28,29]. Despite induction of a typical inflammatory response, reinfection has been observed with ORFV and BPSV [66,67], which suggests that the primary infection fails to confer immunity. This lack of immunity has been partially attributed to the parapoxvirus IL-10, as lesions were substantially reduced during infection and reinfection of sheep with an ORFV in which this gene was deleted [35]. It is, therefore, likely that the other parapoxvirus IL-10s play a similar role during infection and reinfection, but deletion of the BPSV, PCPV, RDPV, and GSPV IL-10 genes in their respective viruses would be necessary to confirm their role in host pathogenesis.

## 4. Materials and Methods

### 4.1. IL-10 Sequences and Bioinformatic Analyses

IL-10 gene sequences from humans, ORFV, BPSV, RDPV, PCPV and GSPV were accessed from GenBank, as indicated in Appendix A. Signal peptides at the *N*-termini of the IL-10 amino acid sequences were predicted using Signal P 5.0 (https://services.healthtech.dtu.dk/service.php?SignalP-5.0, accessed on 9 July 2019). Nucleotide and amino acid sequence lengths were estimated using SnapGene (San Diego, CA, USA). *N*-glycosylation sites within the sequences were predicted using NetNGlyc 1.0 (https://services.healthtech.dtu.dk/service.php?NetNGlyc-1.0, accessed on 15 July 2019).

A multiple sequence alignment was prepared from IL-10 amino acid sequences (lacking signal peptides) using the Clustal Omega tool within Jalview 2.11.0 (http://www.jalview.org/, accessed from 5 August 2019), with the solved crystal structure of hIL-10 as the comparator (PDB identifier 1J7V) [57].

### 4.2. IL-10 Protein Production

Nucleotide sequences for the BPSV, RDPV, PCPV, and GSPV IL-10 genes were modified in SnapGene to include the Kozak translation initiation sequence at the *N*-terminal methionine and the FLAG octapeptide sequence followed by a stop codon at the *C*-terminus, flanked by the restriction enzyme sites for BamHI and XbaI, respectively (Appendix A). Modified genes in the pUC57 plasmid were ordered from GenScript (Piscataway, NJ, USA). The vectors containing the IL-10 genes were digested with BamHI and XbaI (Roche, Basel, Switzerland), then the gene fragments were ligated into the BamHI and XbaI sites generated in the expression vector, pAPEX-3.

HEK-293-EBNA cells were transfected with pAPEX-3-IL-10 constructs using FuGENE^®^-HD reagent (Promega, Madison, WI, USA), then selected using hygromycin B (Gibco, Gaithersburg, MD, USA). Conditioned medium was collected and clarified by centrifugation prior to purification of the FLAG-tagged IL-10 proteins by affinity chromatography with anti-FLAG^®^ M2 affinity resin (Sigma-Aldrich, St. Louis, MO, USA). FLAG-tagged human and ORFV IL-10 were produced in a similar manner, as previously described [33,42].

### 4.3. IL-10 Protein Analyses

To establish concentration, serial dilutions of the purified IL-10 proteins and a control protein (carbonic anhydrase) of known concentration were mixed with SDS-PAGE sample buffer containing 150 mM dithiothreitol (DTT), boiled, then resolved by SDS-PAGE. The proteins were then stained with Coomassie blue, and bands were visualised on the Alliance^TM^ Q9 Advanced Imager (Uvitec, Cambridge, UK) and quantitated using ImageJ [68]. A standard curve obtained for the control protein was used to determine the amount of purified IL-10 protein present in each sample.

To assess dimerisation, IL-10 proteins were mixed with SDS-PAGE sample buffer either with or without 150 mM DTT, without boiling, then resolved by SDS-PAGE. Protein bands were detected by Western blotting with horseradish peroxidase-conjugated anti-FLAG^®^ M2 monoclonal antibody (Sigma-Aldrich). Chemiluminescence was visualised on the Alliance^TM^ Q9 Advanced Imager.

To assess *N*-glycosylation, a Protein Deglycosylation Mix II kit (New England Biolabs, Ipswich, MA, USA) was used to enzymatically remove *N*-glycans from the IL-10 proteins, following the manufacturer’s instructions. Where indicated, proteins (1 µg) were boiled in denaturing buffer followed by incubation with PNGase F at 37 °C for 1 h. Proteins were then resolved by SDS-PAGE and detected by Western blotting, as described above.

### 4.4. Stimulation and Analysis of Pro-Inflammatory Mediator Production in Monocytes

The human THP-1 monocytic cell line was maintained at a density of 1–10 × 10^5^ cells.mL^−1^ in Roswell Park Memorial Institute (RPMI) 1640 supplemented with 10% heat-inactivated foetal bovine serum (FBS), 11 mM glucose, 50 mM 2-mercaptoethanol (2-ME), 1 mM sodium pyruvate and penicillin-G (100 U/mL), streptomycin and kanamycin (100 μg·mL^−1^), with incubation at 37 °C in 5% CO_2_.

To assess anti-inflammatory activity, THP-1 cells (1 × 10^6^ cells·mL^−1^) were stimulated with LTA (1 µg/mL) with or without IL-10 serially diluted in phosphate-buffered saline (PBS). After 24 or 48 h incubation, conditioned media was removed, clarified by centrifugation, then stored at −80 °C. Pro-inflammatory mediators within the conditioned media were then detected using BD-OptEIA^TM^ ELISA kits (IL-8: Cat. No. 555244; MCP-1: Cat. No. 555179; IL-1β: Cat. No. 557953, BD Biosciences, San Diego, CA, USA), following the manufacturer’s instructions.

### 4.5. Stimulation and Analysis of Mast Cell Proliferation

Conditioned media containing IL-3 or IL-4 were prepared from the murine WEHI-3 cell line and concanavalin A-activated murine splenocytes, respectively, as previously described [32]. The murine MC/9 mast cell line was maintained at a density of 2–10 × 10^5^ cells·mL^−1^ in RPMI 1640 supplemented with 10% FBS, 5% IL-4-containing conditioned media, 11 mM glucose, 50 mM 2-ME, 1 mM sodium pyruvate and penicillin-G (100 U/mL), streptomycin and kanamycin (100 μg·mL^−1^), with incubation at 37 °C in 5% CO_2_.

To assess stimulatory activity, MC/9 cells (1 × 10^4^ cells·mL^−1^) were treated with IL-4- or IL-3-containing conditioned media (1.25%) with or without stimulation by IL-10 serially diluted in PBS. After 48 h incubation, the cells were incubated with 3-(4,5 dimethylthiazol-2-yl)-2,5 diphenyl tetrazolium bromide (MTT, 1 mg·mL^−1^, Sigma-Aldrich) for 2 h. Formazan crystals were then solubilised in 10% SDS (in 0.01N HCl, pH 7.4), with incubation in the dark at 37 °C overnight. Absorbance was read at 570 nm then converted to cell number using a standard curve generated by manual cell counting using a haemocytometer. The percentage change in cell number was calculated relative to the difference between IL-3- and IL-4-treated cells with and without hIL-10.

### 4.6. Analysis of hIL-10R1 Binding

To assess hIL-10R-1 binding, a competitive displacement ELISA was used. Maxisorp 96-well immunoplates (Nunc, Roskilde, Denmark) were incubated overnight with 400 ng·mL^−1^ of hIL-10 in coating buffer (15 mM Na^2^CO_3_, 35 mM NaHCO_3_, pH 9.6) at 4 °C. Plates were washed then blocked with 10% bovine serum albumin in PBS for 1 h. Samples of IL-10 serially diluted in PBS containing 0.4% BSA and 0.02% Tween20 were incubated with 300 ng·mL^−1^ hIL-10R1–Fc chimeric protein (R&D Systems, Minneapolis, MN, USA) in non-absorbent plates at 25 °C for 1 h. The mixture was transferred to plates coated with hIL-10 and incubated at 25 °C for 1 h to capture the unbound hIL-10R1–Fc protein. The captured hIL-10R1–Fc protein was detected with a goat anti-human IgG-HRP conjugate (Thermo Fisher Scientific, Waltham, MA, USA) and 3,3′,5,5′-tetramethylbenzidine substrate (BD Biosciences). The reaction was stopped with H_2_SO_4_ and quantified by measuring the absorbance at 450 nm. The percentage change in hIL-10R1-Fc protein bound to the immobilised hIL-10 was calculated relative to that detected in the absence of soluble IL-10.

### 4.7. Data and Statistical Analyses

Concentration–response parameters were obtained by fitting three-parameter nonlinear regression curves (log10 transformed concentration (agonist or inhibitor) versus response-variable slope) using GraphPad Prism v9.0 (GraphPad Software, Inc., San Diego, CA, USA). Data presented in figures and tables are the mean (±SEM or SD) from independent experiments.

Statistical tests were also performed with GraphPad Prism v9.0. One-way ANOVA was applied to pIC50, pEC50 and Emax data, after normality and equality of variance assumptions were verified with Shapiro–Wilk and Brown–Forsythe tests, respectively. Where ANOVA results indicated differences in these parameters between IL-10 treatments, means were compared with a post hoc Tukey test. *p*-values of ≤0.05 were considered significant, as indicated in tables.

## 5. Conclusions

This study provided the first biochemical and functional comparison of five IL-10-like proteins encoded by members of the parapoxvirus genus. The recombinantly-produced parapoxvirus IL-10s varied in their amino acid sequence, molecular weight, and glycosylation state, which translated to differences in their receptor binding, anti-inflammatory and immunostimulatory activities. These analyses yielded mechanistic insight into how hIL-10 mediates its many biological effects, while providing a strong indication as to the virulence role that each parapoxvirus IL-10 plays during infection of their respective hosts.

## Figures and Tables

**Figure 1 pathogens-11-00507-f001:**
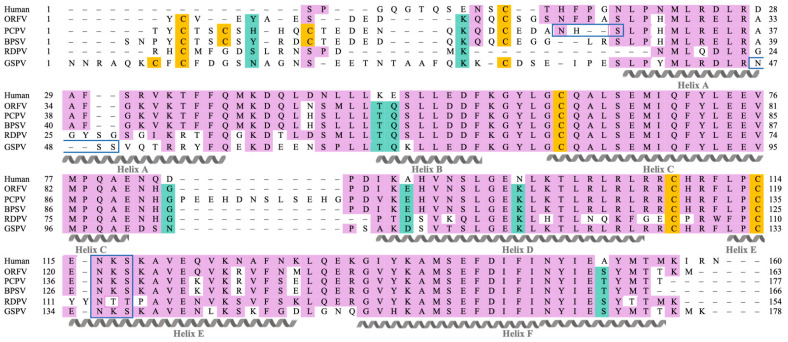
Structural alignment of human and parapoxvirus IL-10s. Multiple sequence alignment of IL-10s from human, ORFV, PCPV, BPSV, RDPV and GSPV (minus predicted signal peptides). Regions of amino acid similarity between human IL-10 and the PPV IL-10 sequences, and between all PPV IL-10 sequences, are highlighted in purple and green, respectively. Alpha helices of hIL-10 are indicated below the alignment and labelled from A to F. Cysteine residues are highlighted in yellow. Sites predicted to show *N*-linked glycosylation are indicated by blue borders.

**Figure 2 pathogens-11-00507-f002:**
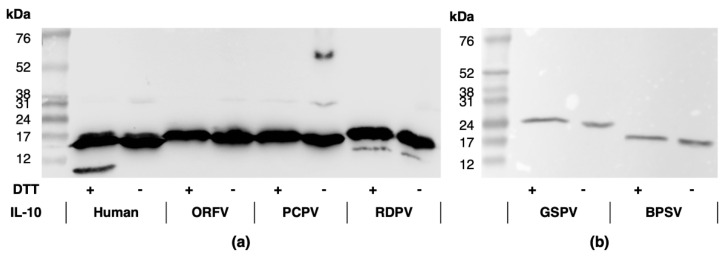
Anti-FLAG Western blot showing bands corresponding to the human, ORFV, PCPV and RDPV (**a**), and BPSV and GSPV (**b**) IL-10 proteins. Molecular weight markers and the presence (+) or absence (−) of reducing agent (DTT) are indicated.

**Figure 3 pathogens-11-00507-f003:**
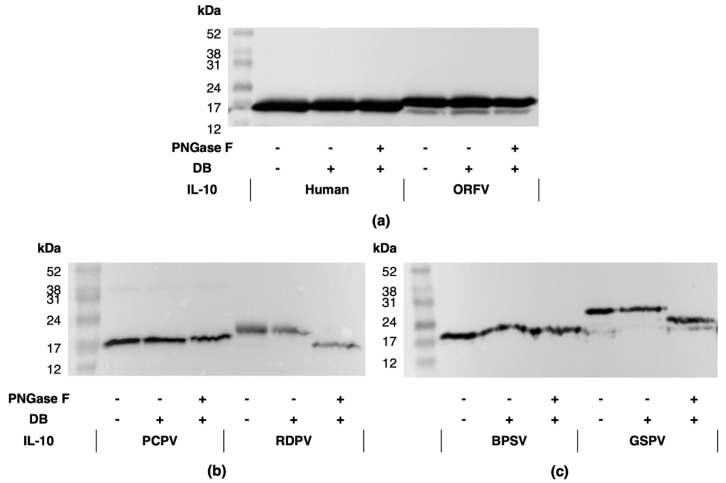
Anti-FLAG Western blot showing bands for human and ORFV (**a**), PCPV and RDPV (**b**), and BPSV and GSPV (**c**) IL-10 proteins following *N*-glycosidase digestion. Molecular weight markers and the presence (+) and absence (−) of denaturing buffer (DB) and PNGase F are indicated.

**Figure 4 pathogens-11-00507-f004:**
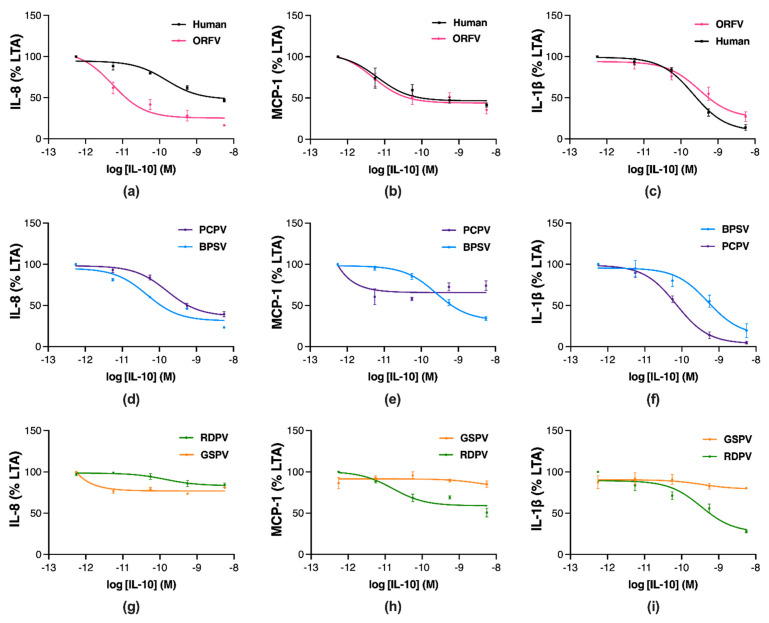
Parapoxvirus IL-10s vary in their inhibition of pro-inflammatory chemokine and cytokine production in stimulated human monocytes. THP-1 cells were stimulated with LTA (1 µg·mL^−1^) and concurrently treated with IL-10s from human ((**a**–**c**), black), ORFV ((**a**–**c**), pink), BPSV ((**d**–**f**), blue), PCPV ((**d**–**f**), purple), RDPV ((**g**–**i**), green), and GSPV ((**g**–**i**), orange). After 24, 24, or 48 h incubation, ELISA was used to assess the levels of IL-8 (**a**,**d**,**g**), MCP-1 (**b**,**e**,**h**), IL-1β (**c**,**f**,**i**), respectively. Non-linear regression plots show the results as a percentage of the cytokine/chemokine levels relative to LTA-stimulated cells, with values representing the mean ± SEM of three independent experiments.

**Figure 5 pathogens-11-00507-f005:**
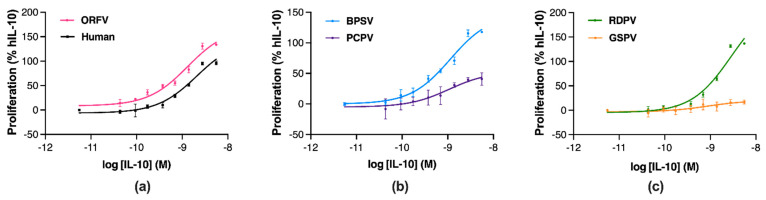
Parapoxvirus IL-10s vary in their stimulation of murine mast cell proliferation. MC/9 cells were co-cultured with IL-4- and IL-3-containing conditioned medium (from concanavalin A-activated murine splenocytes and murine WEHI-3 cells, respectively) and concurrently treated with IL-10s from human ((**a**), black), ORFV ((**a**), pink), BPSV ((**b**), blue), PCPV ((**b**), purple), RDPV ((**c**), green), and GSPV ((**c**), orange). After 48 h incubation, the cell number was assessed through tetrazolium dye reduction and a standard curve. Non-linear regression plots show the results as a percentage of the maximal proliferative response to hIL-10 stimulation, with values representing the mean ± SEM of three independent experiments.

**Figure 6 pathogens-11-00507-f006:**
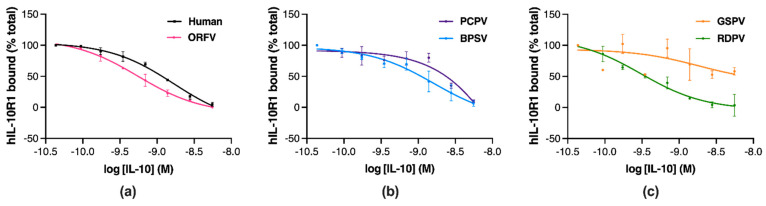
Parapoxvirus IL-10s vary in their binding to hIL-10R1. Soluble hIL-10R1-Fc fusion protein was incubated with IL-10s from human ((**a**), black), ORFV ((**a**), pink), BPSV ((**b**), blue), PCPV ((**b**), purple), RDPV ((**c**), green), and GSPV ((**c**), orange). The mixture was then added to hIL-10 coated wells to capture free hIL-10R1-Fc, which was detected with a biotinylated anti-human Ig-streptavidin conjugate. Non-linear regression plots show the results as a percentage of the maximal absorbance of hIL-10R1-Fc bound, with values representing the mean ± SD of two independent experiments.

**Table 1 pathogens-11-00507-t001:** Potency and efficacy of human and parapoxvirus IL-10s at inhibiting pro-inflammatory chemokine and cytokine production by stimulated human monocytes.

IL-10	IL-8		MCP-1		IL-1β	
	pIC50 (M) ± SEM	Emax (Δ%) ± SEM	pIC50 (M) ± SEM	Emax (Δ%) ± SEM	pIC50 (M) ± SEM	Emax (Δ%) ± SEM
Human	9.79 ± 0.15 ^b^	100 ± 1.94 ^b–f^	11.22 ± 0.47 ^c^	100 ± 2.57 ^d,f^	9.66 ± 0.01	100 ± 2.33 ^e–f^
ORFV	11.17 ± 0.29 ^a,d,e^	156.95 ± 1.60 ^a,c–f^	11.14 ± 0.39	109.89 ± 4.82 ^d–f^	9.50 ± 0.16	84.58 ± 3.85 ^d,f^
BPSV	10.37 ± 0.08	143.93 ± 1.15 ^a–b,d–f^	9.64 ± 0.12 ^a^	112.28 ± 2.20 ^d–f^	9.41 ± 0.23	96.83 ± 6.69 ^f^
PCPV	9.87 ^b^ ± 0.05	113.97 ± 3.40 ^a–c,e,f^	ND	78.09 ± 4.10 ^a–c,f^	10.22 ± 0.08	110.44 ± 1.06^b,e,f^
RDPV	9.74 ^b^ ± 0.29	30.54 ± 3.16 ^a–e^	10.70 ± 0.27	84.21 ± 5.14 ^b,c,f^	9.60 ± 0.34	83.57 ± 1.10 ^a,d,f^
GSPV	ND	50.07 ± 0.93 ^a–e^	ND	31.03 ± 2.78 ^a–e^	11.03 ± 1.88	25.83 ± 1.88 ^a–e^

ND: not able to be accurately determined. ^a^
*p* ≤ 0.05 relative to human IL-10; ^b^
*p* ≤ 0.05 relative to ORFV IL-10; ^c^
*p* ≤ 0.05 relative to BPSV IL-10; ^d^
*p* ≤ 0.05 relative to PCPV IL-10; ^e^
*p* ≤ 0.05 relative to RDPV IL-10: ^f^
*p* ≤ 0.05 relative to GSPV IL-10.

**Table 2 pathogens-11-00507-t002:** Potency and efficacy of human and parapoxvirus IL-10s at stimulating murine mast cell proliferation.

IL-10	MC-9 Proliferation	
	pEC50 (M) ± SEM	Emax (Δ%) ± SEM
Human	8.71 ± 0.21	100 ± 4.47 ^b,d–f^
ORFV	8.80 ± 0.9	123.70 ± 0.86 ^a,c,d,f^
BPSV	8.91 ± 0.18	110.37 ± 0.72 ^b,d–f^
PCPV	9.06 ± 0.36	54.11 ± 3.49 ^a–c,e,f^
RDPV	8.57 ± 0.04	126.99 ± 1.65 ^a,c,d,f^
GSPV	9.30 ± 0.80	25.77 ± 0.49 ^a–e^

^a^*p* ≤ 0.05 relative to human IL-10; ^b^
*p* ≤ 0.05 relative to ORFV IL-10; ^c^
*p* ≤ 0.05 relative to BPSV IL-10; ^d^
*p* ≤ 0.05 relative to PCPV IL-10; ^e^
*p* ≤ 0.05 relative to RDPV IL-10: ^f^
*p* ≤ 0.05 relative to GSPV IL-10.

**Table 3 pathogens-11-00507-t003:** Potency and efficacy of hIL-10R1 binding by the human and parapoxvirus IL-10s.

IL-10	hIL-10R1 Binding	
	pIC50 (M) ± SD	Emax (Δ%) ± SD
Human	8.79 ± 0.16	100 ± 1.6
ORFV	9.24 ± 0.14	104.98 ± 0.27
BPSV	8.72 ± 0.47	98.70 ± 2.17
PCPV	7.66 ± 0.45	94.58 ± 1.08
RDPV	9.51 ± 0.31	107.97 ± 5.71
GSPV	ND	ND

ND: not able to be accurately determined.

## Data Availability

All relevant information is contained within the manuscript or the Appendix A.

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
