# Peer review of "Parapoxvirus Interleukin-10 Homologues Vary in Their Receptor Binding, Anti-Inflammatory, and Stimulatory Activities"

_pathogens, 2022, doi:10.3390/pathogens11050507_

Round 1

Reviewer 1 Report

"Parapoxvirus interleukin-10 homologues vary in their receptor
binding, anti-inflammatory, and stimulatory activities" by Amreen Naqash and colleagues is an excellent work on how IL-10 homologues encoded by parapoxviruses act on immune cell models. The Authors evaluate the anti-inflammatory and stimulatory effects of recombinant IL-10s on THP and MC cells via evaluation of cytokines and chemokines linked to inflammation.

Although the work is precise, I strongly advise sticking to proper and current nomenclature of Poxviruses according to "Virus Taxonomy: 2021 Release" by ICTV. Also, the term "Parapoxviridae" should be replaced with the Parapoxvirus genus in the text. 

Additionally, "transcription" should be deleted (line 393).

Overall, I recommend publication of the manuscript after minor revision.

Author Response

  1. Although the work is precise, I strongly advise sticking to proper and current nomenclature of Poxviruses according to "Virus Taxonomy: 2021 Release" by ICTV.

We would like to thank this reviewer for their positive comments regarding the rigour of our work. We also appreciate their alerting us to the latest ICTV guidelines for Virus Taxonomy. We recognise that our naming of Parapoxvirus of red deer (PVRD) and seal parapoxvirus (SPV) were not in fitting with these guidelines, so have revised these to Red deerpox virus (RDPV) and Grey sealpox virus (GSPV), respectively, throughout the manuscript text, tables, figures, and supplementary information.

  1. Also, the term "Parapoxviridae" should be replaced with the Parapoxvirus genus in the text.

In line 62, “Parapoxviridae” has been replaced with “Parapoxvirus genus” to fit with ICTV nomenclature.

  1. Additionally, "transcription" should be deleted (line 393).

Thank you for alerting us to the redundancy in this sentence; “transcription of” has been deleted (from line 396 in the revised manuscript).

Reviewer 2 Report

Naquesh et al describe the functional characterization of a panel of parapoxvirus IL-10 homologs using cell-based assays to define their ability to bind to target receptors and immune modulatory activities. It is a carefully executed and well argued study, and I very much enjoyed reading the manuscript. I only have a couple of minor comments to make to what is a great paper.

Figure 1

The solid boxes around conserved residues are a little confusing, due to the use of smaller boxes within the large major ones. The authors may want to consider a different solution to how to mark the residues that are not conserved within large blocks that are conserved.

Are EC50 values really given in Molar?

Author Response

  1. It is a carefully executed and well argued study, and I very much enjoyed reading the manuscript. I only have a couple of minor comments to make to what is a great paper.

We would like to thank this reviewer for their generous remarks regarding the quality of our manuscript.

  1. Figure 1: The solid boxes around conserved residues are a little confusing, due to the use of smaller boxes within the large major ones. The authors may want to consider a different solution to how to mark the residues that are not conserved within large blocks that are conserved.

We appreciate the reviewer alerting us to the difficulties in interpreting this figure. We have revised the alignment so that the boxes highlighting conserved residues are replaced by coloured backgrounds (in a colour-blind compatible pallet). We believe that this makes it easier to visualise which residues that are not conserved.

  1. Are EC50 values really given in Molar?

We can confirm that the potency of the parapoxvirus IL-10s at inducing mast cell proliferation was calculated as the negative logarithm (p) of the EC50 (pEC50), which is measured in Moles.  To ensure this is clear to readers, we have defined p on line 176 and clarified the data is log10 transformed on line 515.

We have also reviewed these calculations and have confidence that the pEC50 values presented here are correct. As exampled by human IL-10, the log10 protein concentrations tested in this assay were -10.4M (43 pM, 1.56 ng/mL) to -8.25M (5.6 nM, 200 ng/mL), giving a log10 EC50 of -8.71M (1.95 nM, 70 ng/mL) and pEC50 of 8.71M. These concentrations are in fitting with previous work assessing IL-10 stimulation of mast cell proliferation (50 ng/mL, Imlach et al, J Gen Virol, 2022 [ref 32]; 1-100 ng/mL, Wilbers et al, PLoS ONE, 2017 [ref 45]).